Phalangeal joints kinematics during ostrich (Struthio camelus) locomotion

Zhang Rui zhangrui@jlu.edu.cn 1
Ji Qiaoli 1
Luo Gang 1
Xue Shuliang 1
Ma Songsong 1
Li Jianqiao 1
Ren Lei lei.ren@manchester.ac.uk 1 2
1 Key Laboratory of Bionic Engineering, Ministry of Education, Jilin University , Changchun , The People’s Republic of China
2 School of Mechanical, Aerospace and Civil Engineering, University of Manchester , Manchester , United Kingdom
Abdala Virginia
Electronic publication date: 2017 Jan 12
Publication date: 2017
Volume: 5
Electronic Location ID: e2857
Received 2016 Jul 29; Accepted 2016 Dec 2
Copyright: ©2017 Zhang et al.
Copyright year: 2017
Copyright holder: Zhang et al.
License: This is an open access article distributed under the terms of the Creative Commons Attribution License, which permits unrestricted use, distribution, reproduction and adaptation in any medium and for any purpose provided that it is properly attributed. For attribution, the original author(s), title, publication source (PeerJ) and either DOI or URL of the article must be cited.
License URL: https://creativecommons.org/licenses/by/4.0/

Keywords: Ostrich, Toes, Phalangeal joints, Metatarsophalangeal joint, Locomotion

Funding: National Natural Science Foundation of China No. 51275199 51675221 Science and Technology Development Planning Project of Jilin Province of China No. 20140101074JC This study was supported by the National Natural Science Foundation of China (No. 51275199; 51675221), the Science and Technology Development Planning Project of Jilin Province of China (No. 20140101074JC). The funders had no role in study design, data collection and analysis, decision to publish, or preparation of the manuscript.

==============================
The ostrich is a highly cursorial bipedal land animal with a permanently elevated metatarsophalangeal joint supported by only two toes. Although locomotor kinematics in walking and running ostriches have been examined, these studies have been largely limited to above the metatarsophalangeal joint. In this study, kinematic data of all major toe joints were collected from gaits with double support (slow walking) to running during stance period in a semi-natural setup with two selected cooperative ostriches. Statistical analyses were conducted to investigate the effect of locomotor gait on toe joint kinematics. The MTP3 and MTP4 joints exhibit the largest range of motion whereas the first phalangeal joint of the 4th toe shows the largest motion variability. The interphalangeal joints of the 3rd and 4th toes present very similar motion patterns over stance phases of slow walking and running. However, the motion patterns of the MTP3 and MTP4 joints and the vertical displacement of the metatarsophalangeal joint are significantly different during running and slow walking. Because of the biomechanical requirements, osctriches are likely to select the inverted pendulum gait at low speeds and the bouncing gait at high speeds to improve movement performance and energy economy. Interestingly, the motions of the MTP3 and MTP4 joints are highly synchronized from slow to fast locomotion. This strongly suggests that the 3rd and 4th toes really work as an “integrated system” with the 3rd toe as the main load bearing element whilst the 4th toe as the complementary load sharing element with a primary role to ensure the lateral stability of the permanently elevated metatarsophalangeal joint.

Introduction

Ostriches have a large number of adaptations that allow them to move both economically and quickly. They (Struthio camelus) are acknowledged as the fastest and largest extant bipedal land animal also with extraordinary endurance during locomotion and can possibly run faster than antelopes of a comparable size (Schaller et al., 2009; Alexander et al., 1979; Abourachid & Renous, 2000; Schaller et al., 2009; Schaller et al., 2011). The ostrich has been filmed running steadily for 30 min at a speed exceeding 50 km/h and moving at a speed of 70 km/h for short sprints, with a step length reaching up to 5 m (Abourachid & Renous, 2000; Schaller et al., 2011). In addition, it is also reported that they are capable of cutting maneuvers with minimal changes of their leg kinematics and joint torques (Jindrich et al., 2007). Some studies showed that ostriches are highly adapted to very economic locomotion from slow walking to fast running (Rubenson et al., 2004; Rubenson et al., 2010).

Ostrich leg morphology may provide the mechanical foundation for this unique locomotor performance (Schaller et al., 2011). For instance, compared to other large cursorial ratite birds, e.g., rhea (Rhea spp.), emu (Dromaius novaehollandiae), cassowary (Casuarius spp.), ostrich has the longest absolute limbs that contribute to achieve great step lengths and step frequency (Gatesy & Biewener, 1991). In addition, the proportion of ostrich hindlimb bones and multi-jointed muscle tendon system are highly adapted for locomotion. On the other hand, compared to other terrestrial birds, such as rhea (Rhea spp.), emu (Dromaius novaehollandiae) and brown kiwi (Apteryx australis), ostriches relatively erect femurs increase the joint chain extension and symmetrical movement (Abourachid & Renous, 2000). Furthermore, the ligaments system and tendons in the hindlimb joints play a vital role in ostrich gaits performance and economy. Ligamentous system of the intertarsal joint prevents tarsometatarsal rotations by providing a primary guiding function and ensuring joint coherence throughout range of motion. During stance phase, the extended intertarsal joint is sustained in the engaged state to provide additional support for body mass (Schaller et al., 2009). The distal part of their hindlimb are primarily controlled by the long and stretched tendons; therefore, the metatarsophalangeal joint may play an important role in storing and releasing elastic energy, and absorbing shock during fast locomotion, hence providing an energy-saving mechanism (Alexander, 1984; Alexander, 1985; Gatesy, 1991; Castanet et al., 2000; Almeida Paz et al., 2008).

It is noteworthy that unique adaptations are also evident in ostrich toe morphology. While ordinary birds have three or four toes, the ostrich has only two toes, the main 3rd toe and the lateral 4th toe. Another unique adaptation at the distal part of the hindlimb is the supra-jointed toe posture with the metatarsophalangeal joint and proximal phalanx of both toes being permanently elevated above the ground surface (Schaller et al., 2011; Deeming, 2003). Pressure plate data suggested that both toes play a vital role in ostrich terrestrial locomotion with different load distributions in walking and running. The 3rd toe sustains most of the ground reaction force during locomotion and its claw provides the forces at push-off in fast locomotion. In addition, the 4th toe functions as a lateral support during locomotion (Schaller et al., 2007; Schaller et al., 2011; Schaller, 2008). The only rigid element in ostrich toes is the single remaining claw that functions as a positional anchor at fast speed when embedded in the terrain (Schaller et al., 2011). The major tendons are distributed in tibiotarsus, tarsometatarsus and two digits (Gangl et al., 2004). There is an interphalangeal ligament inserting mediodistally at the proximal phalanx of the 3rd toe and medioproximally at the second phalanx of the 4th toe (Schaller et al., 2011). This ligament couples the toes motion and limits the 4th toe abduction to the 3rd toe main direction. Therefore, ostrich toes may execute locomotion through the movement coordination with each other and form an “integrated system.”

Although a large number of studies have been conducted to investigate the ostrich hindlimb kinematics during locomotion (Haughton, 1865; Alexander et al., 1979; Alexander, 1985; Gatesy & Biewener, 1991; Abourachid & Renous, 2000; Jindrich et al., 2007; Rubenson et al., 2004; Rubenson et al., 2007; Rubenson et al., 2010; Watson et al., 2011; Smith et al., 2006; Smith et al., 2007; Smith, Jespers & Wilson, 2010; Smith & Wilson, 2013; Schaller et al., 2009; Schaller et al., 2011; Birn-Jeffery et al., 2014; Hutchinson et al., 2015; Rankin, Jonas & Hutchinson, 2016), those kinematic analysis are mainly focused on hip, knee, the metatarsophalangeal and intertarsal joints. So far, little is known about the relative motions of the 3rd and 4th toes intrinsic joints during ostrich foot locomotion.

As the only body part in contact with the ground surface, the unique toe joint motions may play important biomechanical roles during locomotion. Therefore, a better understanding of the interphalangeal joint of toes and the metatarsophalangeal joint kinematics may provide valuable information to reveal the biomechanical mechanism underlying the extraordinary locomotor performance of ostriches. Our primary aims were to test the hypothesis that the 3rd and 4th toes work as an “integrated system,” and motions of the metatarsophalangeal joint and the interphalangeal joints of the 3rd and 4th toes have significantly different patterns during slow walking and running gaits.

In this study, we examined the kinematics of all major joints of ostrich toes in vivo during slow walking and running using high speed videos and specially designed markers. This included the interphalangeal joint motions within both toes, the relative motions between the first phalanx of the 3rd and 4th toe with respect to the tarsometatarsus, and the angle between the first phlanax of the 3rd and 4th toe over entire stance phases. Statistical analysis was also conducted to investigate the effect of locomotor gait on those joint motions. This study aimed to investigate whether there were differences between the two toes joint motions in slow walking and running. To test our hypothesis that phalangeal joint angle and the vertical displacement of the metatarsophalangeal joint at touch-down, mid-stance, lift-off, joint range of motion, maximum and minimum joint angles were selected as key indicators for statistical tests.

Materials and Methods

Animals

Ten healthy sub-adult ostriches (Struthio camelus) with an average age of eight months were selected from the Ji’an breeder, Jilin province, P.R. China. The average mass and height of these ostriches are 84.5 ± 2.12 kg and 2.11 ± 0.01 m (displayed by means ± S.D), respectively. Without any form of surgical treatment or invasive physical manipulation, the individuals were in excellent physical condition with the properly elevated metatarsophalangeal joints, which represented the average body proportion and weight for ostriches of their age and sex (Deeming, 2003). These ostriches were kept in outdoor enclosure in daytime with unlimited access to food and water, and housed in an indoor enclosure at night. Each bird was trained to walk and run on a fenced-in corridor at least 30 min each time, twice per day over a month before data collection. After comprehensive comparison of representation and amenability, two tractable female sub-adult ostriches were selected as objects to complete all tests. All living and experimental conditions were approved by the Institutional Animal Care and Use Committee (IACUC, protocol number: 20140706) of Jilin University, P.R. China.

Experimental setup and trials

A 80 m long runway fenced by 1.5 m tall wire mesh was set up in the breeding field with a data acquisition area in the middle of 4m long and 1 m wide zone (see Fig. 1). The runway was covered with a 3 mm non-slip rubber sheet to prevent potential damage from ostrich foot. At both ends of the runway, large spaces were provided for the ostriches to rest and eat. The area outside of the data acquisition zone was about 76 m long and 2 m wide with two “V” shape transition areas gradually connecting to the data acquisition zone, which helps guide the ostriches to naturally enter into the data acquisition area. A high-speed video system with three synchronized digital cameras (Casio Exilim EX-FH25; Casio, Tokyo, Japan: 240 frames s−1) was placed around the central zone of the data acquisition area in a triangle shape with one camera positioned perpendicular to the sagittal plane of motion (see Fig. 1).

Figure 1 Schematic diagram of the experimental site of 80 m long.

The data acquisition area in the center of the dotted box is of 4 m long and 1 m wide. Fences of 1.5 m high were set on both sides of the runway. Three high-speed cameras were placed in the central data acquisition area in a triangular shape. Both ends of runway are rest areas for ostriches to rest and eat foods.

During measurements, ostriches were led by their breeders or experimenters, using positive reinforcement such as food rewards and vocal commands, with the goal of maintaining a steady speed across a straight distance of about 15 m. Experimenters randomly varied the speed from slow walking to fast running across trials and allowed ample rest and food between trials to prevent fatigue. Experiments were cancelled if animals showed fatigue that would cause discomfort or adversely affect our measurements. To minimize the interference of sunlight, one sunshade net was set on the top of the data acquisition zone.

Marker placements and joint angles

Nine specially designed thermoplastic plates carrying nine retro-reflective markers were firmly mounted at the major anatomical landmarks around the ostrich left foot toes using double sided tapes (see Fig. 2A). The marker locations were determined by palpation and referring to a three-dimensional (3D) geometric model of the tarsometatarsus bone and the phalanges of the 3rd and 4th toes, reconstructed from the CT images of a healthy adult female ostrich (Age: 3 years, Weight: 95 kg, Height: 2.10 m) left foot by using Mimics 10.0 software (Materialise, Leuven, Belgium) (see Fig. 2B). Four markers were used for the 3rd toe at the dorsal base of the toe nail (marker A), the joint between phalanges II and III (marker B), the joint between phalanges I and II (marker C), and the joint between phalanx I and tarsometatarsus (marker D), whereas, three markers were placed on the 4th toe at the joint between phalanx I and tarsometatarsus (marker F), the joint between phalanges I and II (marker G), and the distal end of the 4th toe (marker H). Additionally, one marker was put on the anterior side of the tarsometatarsus bone proximal to the metatarsophalangeal joint (marker E). Here, the toe nail, phalanges III and IV of the 3rd toe were considered as one segment (phalanges III in Fig. 2B), and the phalanges II III IV and V of the 4th toe were assumed as one part (phalanges II in Fig. 2B) because these phalanges are small and the relative motions among them are hard to measure and observe (Fowler, 1991).

Figure 2 The reflective markers on ostrich foot and the toe joint angle measured.

Nine reflective markers were placed at the major anatomical landmarks of ostrich toes (A). The marker locations were determined by palpation and referring to a 3D geometric toe model reconstructed from the CT images of a healthy adult female ostrich (Age: 3 years, Weight: 95 kg, Height: 2.10 m) left foot (B). Six toe joint angles were defined (C): angle α between the phalanges II and III of the 3rd toe, angle β between the phalanges I and II of the 3rd toe, angle γ between the tarsometatarsus and the phalanx I of the 3rd toe (MTP3 joint), angle θ between the tarsometatarsus and the phalanx I of the 4th toe (MTP4 joint), angle ϕ between the phalanges I and II of the 4th toe, angle ψ was between the first phalanges of the 3rd and 4th toes and all angles were spatial 3D angles. MTP3 represents the joint between tarsometatarsus and the phalanx I of the 3rd toe. MTP4 represents the joint between the tarsometatarsus and the phalanx I of the 4th toe. MTP represents the metatarsophalangeal joint. MTP joint include the MTP3 and MTP4 joints.

The 3D coordinates of the nine retro-reflective markers were measured at 240 Hz using a three-camera (Casio Exilim EX-FH25; Casio, Tokyo, Japan) motion tracking system (Simi Motion 2D/3D® 7.5 software; SIMI Reality Motion Systems GmbH, Germany). Two series of representative video frames recorded for slow walking and running respectively are shown in Fig. 3. The marker data and joint kinematics were analyzed using Simi Motion 2D/3D® 7.5 software. The software allows for three-dimensional calibration, digitization of bony landmarks and calculation of the segment and joint kinematic parameters of interest (Stoessel & Fischer, 2012). The average error of motion tracking system measurement accuracy was ±1.0 mm. The time histories of six joint angles were calculated; namely, angle α between the phalanges II and III of the 3rd toe, angle β between the phalanges I and II of the 3rd toe, angle γ between the tarsometatarsus and the phalanx I of the 3rd toe (MTP3 joint), angle θ between the tarsometatarsus and the phalanx I of the 4th toe (MTP4 joint), angle ϕ between the phalanges I and II of the 4th toe, and angle ψ between the first phalanges of the 3rd and 4th toes (see Fig. 2C). Also, we measured displacements of the metatarsophalangeal joint z and directly exported data by motion tracking system.

Figure 3 Two representative high speed video traces of toe motions during slow walking and running in stance phases.

The traces started at touchdowns when the 3rd toe touched the ground at 0% of stance phase. In the slow walking and running trials, the mid-stance is at 50% of stance phase and the 3rd toe cleared off the ground at 100% of stance phase.

Animal forward average velocity was calculated by stride length divided by stride period. We defined steady state trials as those in which the absolute difference between the forward velocities at two consecutive touch downs was less than 20% of the average forward velocity. Trials with greater or smaller values of acceleration/deceleration were discarded. Froude numbers Fr = v2∕(gh) and dimensionless speed (u = Fr0.5) were calculated to normalize speeds, where v is the forward velocity of the animal, g is the gravitational-acceleration constant and h is the length of the pendulum (leg length from hip to ground) (e.g., Alexander & Jayes, 1983). Gait parameters, including, cycle period, stance duration, swing duration, duty factor and stride length were calculated for each steady state trial.

Statistical analysis

Statistical analyses were conducted to examine the differences in four gait parameters (stance and swing duration, cycle period and stride length), six key indicators (phalangeal joint angle/the vertical displacement of the metatarsophalangeal joint at touch-down, mid-stance, lift-off, maximum, minimum and range of motion) between slow walking and running gaits using Origin Pro 2015 software (OriginLab Corporation, Northampton, MA, USA). In this study, trials with stance duration > 0.9 sand duty factor >  0.5 were considered as walking gaits at slower speeds with double support, whereas trials with stance duration < 0.9 sand duty factor < 0.5 were regarded as running gaits (Schaller et al., 2011). We used one-way ANOVA statistical technique to analyze the effect of locomotor gait on each gait parameter or joint angle/displacement indicator (Schache et al., 2011; Stoessel & Fischer, 2012). Using the F-test to test whether these two variations are significantly different. Statistical significance level was considered as P < 0.05.

An equal number of stance phases were included in the statistical analysis from each individual for both slow walking and running in order to weight each evenly. Additionally, in order to study the potential for inter-subject variation, interphalangeal joint angle values of slow walking and running trials from each individual were conducted to an analysis of variance. A total of 38 samples (individual A, 19 samples; individual B, 19 samples), divided between slow walking and running trials, were included in the statistical analysis (see Table 1).

Table 1 The statistical analysis trials number of two individuals during slow walking and running gaits.

	Slow walking trials	Running trials	
Individuals	A	B	A	B	
Valid stance phase	15	12	7	9	
Statistical analysis trials	12	12	7	7	

Results

Gait parameters

Averages and standard derivations of key gait parameters, including stance duration, swing duration, cycle period and stride length of all slow walking and running gaits were listed in Table 2 separately. It can be seen that there were statistically significant differences in stance duration, cycle period and stride length between slow walking and running gaits. Ostriches use considerably shorter cycle periods and stance duration during running than those during slow walking, whereas dramatically increase their stride lengths (Abourachid & Renous, 2000). There was no statistically significant difference was found in swing duration between slow walking and running gaits. These observation were consistent with previous observation (Alexander et al., 1979; Rubenson et al., 2004).

Table 2 The key gait parameters during slow walking and running gaits.

Gait parameters	Slow walking (0.38–1.23 m/s)	Running (2.26–3.31 m/s)	
Number of trials	56	25	
Statistical analysis stance phases	24	14	
Average speed (m/s)	0.84 ± 0.20*	2.77 ± 0.28*	
Froude numbers	0.06 ± 0.03*	0.66 ± 0.13*	
Duty factor	0.74 ± 0.09*	0.45 ± 0.03*	
Stance phase (s)	1.22 ± 0.33*	0.34 ± 0.03*	
Swing phase (s)	0.44 ± 0.16	0.42 ± 0.02	
Cycle period (s)	1.66 ± 0.30*	0.76 ± 0.03*	
Stride length (m)	1.33 ± 0.16*	2.11 ± 0.15*	
Notes.

Values are means ± S.D.

* indicates statistically significant speed effects (P < 0.05).

Toe joint kinematics

Figure 4 showed the averages and one standard deviation zones of the six toe joint angles and the vertical displacements of metatarsophalangeal joint (α, β, γ, θ, ϕ, ψ, z) over the stance phases for all slow walking and running trials respectively. From Fig. 4A, it can be seen that the time trajectories of the third phalangeal joint angle of the 3rd toe (α) shared very similar patterns in the stance phases during slow walking and running. The third phalangeal joint of the 3rd toe extended about 10° immediately after the touch-down, and thereafter remained at about 165° throughout from early stance to late stance. This was followed by a swift flexion of about 35° and also a rapid extension of about 20° just before lift-off. However, compared to slow walking gaits, it appeared that during running the joint extension in the early stance finished slightly later (at 20% of the stance phase), and the joint flexion and protraction in the late stance occurred earlier (at 70% of the stance phase).

Figure 4 The averages and one standard deviation zones of the six toe joint angles and the vertical displacement of the metatarsophalangeal joint (α, β, γ, θ, ϕ, ψ, z) (corresponding to A B C D E F G, respectively) over the stance phases for all slow walking (blue dotted line) and running trials (red solid line). Angle decrease represents flexion while angle increase indicates extension. The stick figures at the bottom of (H) and (I) showed the ostrich foot motion in stance phase and the green point indicates the metatarsophalangeal joint.

From Fig. 4B, we could see that the second phalangeal joint angle of the 3rd toe ( β ) showed similar patterns in the stance phases of slow walking and running. The joint angle decreased after touch-down from about 160° to 120° or so at late stance. Thereafter, the joint extended swiftly back to about 160° just before lift-off. However, it is noteworthy that the joint flexed and extended much more radically in running than during slow walking, and high variability was observed from early stance to middle stance (from touch-down to 60% of stance phase).

The MTP3 joint angle γ was been shown in Fig. 4C present noticeably different patterns during slow walking from running. In slow walking, the joint angle decreased gradually from touch-down to 30% of stance phase about 25°, and then remained steady till reaching 80% of stance phase. A sharp joint angle increased occurs just before lift-off with the MTP3 joint extended almost 60° whereas in running there was no plateau stage in middle stance. The joint flexed gradually from touch-down to middle stance, and thereafter extended progressively to 210° at lift-off.

The MTP4 joint angle θ also showed different motion patterns during slow walking from running (see Fig. 4D). In slow walking, the MTP4 joint flexed about 25° directly after touch-down, and remained reasonably steady from early stance through to late stance. Just before lift-off, a swift joint extension occurred at the MTP4 joint reaching a nearly fully extended position at 170° whereas, in running gaits, there was no steady stage in the middle of stance phase. After touch-down, the MTP4 joint flexed gradually about 30° till middle stance, and thereafter followed by a progressive joint extension of 60° till lift-off.

The largest angle variability among all the six toe joints was observed at the first phalangeal joint angle of the 4th toe (ϕ). From Fig. 4E, we can see that no apparent patterns presented for angle ϕ during both slow walking and running. The joint angle fluctuated around 165° though it appeared that larger variability occurred during slow walking rather than running. While angle (ψ) between the first phalanges of 3rd and 4th toes showed clear patterns over the stance phase (see Fig. 4F). The angle between the two toes moved similarly during slow walking and running with a gradually increasing the 4th toe abduction to the 3rd toe main axis from touch-down to late stance followed by a swift adduction before lift-off. The average peak joint extension was about 39° for both slow walking and running. This is nearly consistent with previous study that the maximum motion range angle between the 3rd and 4th toe main axes was 34° (Schaller et al., 2011).

Figure 4G showed the average and one standard deviation zone of the vertical displacements of the metatarsophalangeal joint (z) over stance phases for all slow walking trials and for all running trials respectively. It can be seen that markedly different patterns were present during slow walking compared to running. In slow walking, the metatarsophalangeal joint moved downwards towards the ground surface about 3.0 cm just after touch-down, and thereafter went smoothly upwards about 20 cm before lift-off whereas, during running, the joint only moved downwards slightly about 6.2 cm from touch-down to near middle stance, and then kept going upwards before lift-off about 20 cm. This was nearly consistent with Figs. 4H and 4I that displayed motion trajectory of ostrich foot and the metatarsophalangeal joint during slow walking and running gaits. We can see that the motion trajectory during running gait was more smooth than that in slow walking gait.

Figure 5 showed that the six toe joint angles and the vertical displacement of the metatarsophalangeal joint had similar motion patterns over stance phase during slow walking and running gaits between individual A and individual B.

Figure 5 The averages and one standard deviation zones of the six toe joint angles and the vertical displacement of the metatarsophalangeal joint (α, β, γ, θ, ϕ, ψ, z) over the stance phases for all slow walking and running trials.

Individual A were shown in blue dotted line and individual B were shown in red solid line.

Figure 6 showed that no statistically significant inter-individual differences were found in the six toe joint angles and the vertical displacement of the metatarsophalangeal joint at touch-down, mid-stance, lift-off and also the ranges of motion during slow walking and running gaits.

Figure 6 The averages and standard deviations of the six toe joint angles and the vertical displacement of the metatarsophalangeal joint at touch-down, mid-stance, lift-off and also the ranges of motion during slow walking and running between individual A and individual B.

Statistically significant effect of individual difference are indicated by an asterisk (P < 0.05). No significant inter-individual effects were found, since there is no * in the figure.

Effect of locomotor gait

The results of the statistical analysis examining the effect of locomotor gait on the six key indicators (angles/displacements at touch-down, mid-stance, lift-off, maximum, minimum and range of motion) of the six toe joint angles and the vertical displacement of the metatarsophalangeal joint were listed in Fig. 7 and Table S1. Among all the six toe joints, the MTP3 and MTP4 joints showed the largest ranges of motion whereas, angle ϕ had the smallest range of motion. As shown in Fig. 7, no statistically significant differences were found for the six key indicators of angles α, ψ between slow walking and running gaits. Statistically significant differences were found for the range of motion of the second phalangeal joint angle of the 3rd toe (β) and the lift-off angle of the first phalangeal joint angle of the 4th toe (ϕ). A slightly larger range of motion of the second phalangeal joint angle of the 3rd toe (β) presented during slow walking than running. In addition, the first phalangeal joint angle of the 4th toe (ϕ) flexed much more at lift-off during running than slow walking.

Figure 7 The averages and standard deviations of the six toe joint angles and the vertical displacement of the metatarsophalangeal joint at touch-down, mid-stance, lift-off and also the ranges of motion during slow walking and running.

Statistically significant effect of locomotor gait are indicated by an asterisk (P < 0.05).

Figure 7 shows that statistically significant differences were found in several key indicators of the MTP3 joint angle (γ), the MTP4 joint angle (θ) and also the vertical displacement (z) of the metatarsophalangeal joint between slow walking and running trials. This was consistent with the distinct patterns we observed in Fig. 4. In addition, the MTP3 joint flexed much more at touch-down, mid-stance, and used a larger range of motion during running compared to slow walking. The MTP4 joint presented a more flexed positions at touch-down and mid-stance, and a more extended position at lift-off during running. This led to a larger range of motion at the MTP4 joint in running trials. Therefore, the MTP joint had a greater range of motion during running than that in slow walking, which may further explain the metatarsophalangeal joint played an important role as the energy storage and shock absorption during fast locomotion (Schaller, Herkner & Prinzinger, 2005; Rubenson et al., 2007). For the vertical displacement of the metatarsophalangeal joint, although very similar ranges of motion were used during slow walking and running, the metatarsophalangeal joint was at a statistically higher position at mid-stance during slow walking, which was consistent with the viewpoint that the metatarsophalangeal joint was positioned closer to the ground, as speed increased (Schaller et al., 2009).

Discussions

This study first presents toe joint kinematic analysis in sub-adult ostriches during overground slow walking and running. Reliable data on major toe joint angle trajectories and metatarsophalangeal vertical displacement were obtained base on a large number of trials, allowing interpretation of toe function in this flightless, cursorial bird with a unique elevated metatarsophalangeal joint supporting only by two toes. Having chosen two genetically unrelated subjects of the same sex and very similar age and size, the consistency of inter-individual results in slow walking and running trials accurately document a generalized pattern in ostrich locomotion. However, ostrich maturity has not been considered in our study. Some studies suggested that scaling of kinematic variables largely agreed with predicted scaling for increasing size. This demonstrated that there was close relationship for dynamic similarity between sub-adult and adult ostriches (Smith, Jespers & Wilson, 2010). Ontogenetic scaling of locomotor mechanics largely resulting from simple scaling of the limb segments rather than postural changes.

Ostrich toes might play essential role in force and power generation, and also energy saving for slow walking and running gaits (Schaller et al., 2009; Schaller et al., 2011). Our study reveals that almost all the six major toe joints present notably large motions from slow to fast locomotion. The MTP3 and MTP4 joints exhibit the largest range of motion among all the six toe joints with an average range of motion about 70° in slow walking and a higher motion range of 80° during running. The smallest range of motion is found at the first phalangeal joint of the 4th toe, but still has an average range of about 30°. Rough skins, sturdy ligaments, fascia and lumpy fat pads envelop the metatarsophalangeal joints, toe skeleton and interphalangeal joints to ensure structural integrity, seemingly providing limitation on toe mobility (Schaller et al., 2011). In addition, the lower hindlimb and multi-jointed muscle tendon system are primarily activated by long tendons that store and release elastic energy during fast locomotion to provide an energetic advantage (Schaller et al., 2011). The unique posture of the supra-jointed metatarsophalangeal joint elastic energy storage structures is primarily maintained by ligaments (Schaller et al., 2009). The metatarsophalangeal joint likely stores and releases elastic energy during fast locomotion than slow walking gait.

Two toes as an “integrated system”

The 3rd toe and claw essentially forming an extension of the tarsometatarsal limb sustain most of the impact force at touch-down and ensure stable load bearing and grip during stance phase (Schaller et al., 2011). During slow walking and running, just after touch-down, simultaneous flexions at the first phalangeal joint of the 3rd toe and the MTP3 joint, and also an extension at the second phalangeal joint of the 3rd toe occurs implying compliance at the interphalangeal joints of the 3rd toe is used to moderate ground impact at touch-down. Thereafter, the second phalangeal joint remains fully extended in contact with the ground surface whereas the first phalangeal joint flexes gradually till late stance. Interestingly, the third phalangeal joint angle of the 3rd toe (α) and the second phalangeal joint angle of the 3rd toe (β) show statistically very similar motion patterns during running compared to slow walking. Since no intrinsic muscles exist in ostrich toes (Gangl et al., 2004), this suggests that the tensions are well tuned by at the toe flexors and extensors tendons crossing the different joints come from the same digital flexor muscle during running, not only to counteract the higher ground reaction forces but also to regulate the interphalangeal joint motions (Schaller et al., 2011).

The first phalangeal joint of the 4th toe presents the largest motion variability among all the six toe joints examined in this study with no obvious patterns found during slow walking and running. This appears to support the previous hypothesis proposed by Schaller that the 4th toe acts as a stabiliser to compensate uneven ground surface and adjust potential body imbalance (Schaller et al., 2011). This seems further supported by the results that the deviation of the motion range of the the first phalangeal joint angle of the 4th toe (ϕ) during slow walking is almost twice higher than that of running. Slow moving may need a greater level of neural control and muscular regulation of the ground contact elements (Kummer, 1959; Schaller et al., 2011). The angle between the first phalanges of the 3rd and 4th toes (ψ) shows very similar pattern during slow walking and running implying the high stiffness of the interphalangeal ligaments connecting the 3rd and 4th toes. Besides that, in order to measure and observe the 4th phalangeal joint motion, we considered the phalanges II, III, IV and V of the 4th toe as one segment. However, there may be some smaller motions within this simplified segment as well as a high sensitivity to marker placement. Therefore, this is probably one of the reasons that the first phalangeal joint of the 4th toe shows the largest motion variability. Over most of the duration when the 4th toe is in contact with the ground during slow walking, the average angle between the first phalanges of the 3rd and 4th toes (ψ) is only about 25°. This is much lower than the maximum angle (34°) determined by a fresh anatomical dissection study (Schaller et al., 2011) and also the in vivo maximum value (46 ± 8°) recorded in this study. The average angle between the first phalanges of the 3rd and 4th toes (ψ) further drops to about 20° during running. Schaller et al. suggested that the 4th toe presumably allows compensation for uneven ground conditions to correct potential imbalances in CoM (center of mass), particularly at slower speeds (Schaller et al., 2011). In addition, the significantly lower variation in load distribution when running illustrates the effects of dynamic stability, which reduces the demand for fine adjustment at the 4th toe (Schaller et al., 2011). This strongly suggests that the body stabilization function of the 4th toe due to its lateral orientation may be as pronounced as proposed by the previous study especially for fast locomotion (Schaller et al., 2011). The major function of the 4th toe might be to offset the ground impact and reaction forces during early and middle stances, thereby to provide extra support for the lateral stability of the elevated metatarsophalangeal joint as the body weight transfers laterally.

Although the interphalangeal joints of the 3rd and the 4th toes present distinct motion patterns in stance phases of slow walking and running, the two major joints (MTP3 and MTP4 joint) connecting the two toes to the tarsometatarsus share highly similar patterns for both slow and fast locomotion (see Figs. 4C and 4D). The average trajectories of the MTP3 joint angle (γ) and the MTP4 joint angle (θ) are almost perfectly in phase over the entire stance phases. This strongly suggests that the 3rd and 4th toes actually move as an “integrated system” from slow to fast locomotion. This synchronous pattern is more pronounced during running when the 4th toe lies more closely to the 3rd toe due to a smaller average angle between the 3rd and 4th toes (ψ) during most of stance phase. Moreover, from our high speed videos of running trials, we found that after the 4th toe clears off the ground, it aligns in a line and almost forms a single segment with the 3rd toe at push-off. This highly concerted toe motion is probably an emerging result of the dynamic interaction of the proximal leg musculature, the distal passive ground contact apparatus and the external environment. The leg muscles of ostriches are highly concentrated at the proximal joints resulting in reducing moment of inertia with respect to the proximal joints. This enables ostriches to achieve high step frequency energy efficiently (Schaller et al., 2011; Haughton, 1865). The permanently elevated metatarsophalangeal joint further increases the leg length thereby leading to higher stride length (Rubenson et al., 2007; Schaller et al., 2011). Even though no intrinsic muscles are present to delicately regulate the toe motions (Gangl et al., 2004), our toe joint motion data strongly suggests that the toe joints are appropriately controlled by well-tuned tensions at toe flexor and extensor tendons. Indeed, the ostrich intertarsal joint can be considered as a passive mechanism well regulated by distal limb tendons and ligaments to work as an “integrated system” to generate ground reaction forces, attenuate ground impacts and accommodate ground surfaces whilst ensuring the stability of the elevated metatarsophalangeal joint from slow to fast locomotion.

Different strategies at slow and fast locomotion

Our measurement data indicates that significantly different time history patterns are used by ostriches in the vertical displacement of the metatarsophalangeal joint and also the joint motions at MTP3 and MTP4 joints during running compared to slow walking. This is supported by the statically significant differences found in a number of key indicators of the displacement z, the MTP3 joint angle (γ) and the MTP4 joint angle (θ). In most of stance phase of slow walking (10%–80%), the metatarsophalangeal joint only moves slightly upwards mainly due to the flexion of the first phalangeal joint of the 3rd toe because both the MTP3 and MTP4 joints and also the second phalangeal joint of the 3rd toe remain almost stationary over this period.

Previous study revealed that ostriches used an inverted pendulum gait at slow locomotion (Rubenson et al., 2004). The out-of-phase pattern in the fluctuations of the potential and kinetic energies allows for a high percentage of mechanical energy recovery at slow speeds, which are typical of walking gait in bipedal species (Cavagna, Thys & Zamboni, 1976; Cavagna, Heglund & Taylor, 1977; Heglund, Cavagna & Taylor, 1982; Muir, Gosline & Steeves, 1996); whereas, at fast locomotion (including grounded running and aerial running), ostriches tend to use a bouncing gait by using the legs as a springy mechanism to store and regain energy characterized by a marked reduction in the phase difference between the potential and kinetic energies (Haughton, 1865; Alexander et al., 1979; Muir, Gosline & Steeves, 1996; Rubenson et al., 2004; Daley & Aa, 2006; Jindrich et al., 2007; Rubenson et al., 2010; Schaller et al., 2009; Schaller et al., 2011; Andrada et al., 2013; Birn-Jeffery et al., 2014; Hutchinson et al., 2015; Rankin, Jonas & Hutchinson, 2016). The distinct toe joint motions at slow and fast locomotion observed in this study are probably the direct result of the selective use of those two distinct energy strategies at different speed ranges. At low speeds, the metatarsophalangeal joint only moves slightly over most of the stance phase (10%–80%) by mainly using the first interphalangeal joint motion at the 3rd toe. However, at high speeds, the metatarsophalangeal joint presents a typical loading and rebounding pattern over the stance phase by mainly using the significant flexion and extension motions at the MTP3 and MTP4 joints possibly regulated by the stretched toe flexor tendons. This suggests that at fast locomotion the toes also work as a springy element in series with the proximal limb to attenuate ground impact, generate supporting forces and also may save metabolic energy cost.

In addition, in our study ostrich running speed was much lower than 50 km/h (−13 m/s), thus toe-joint motion may be different from the result of average speed 2.77 m/s. This is mainly because locomotion was initiated either by display of food at the end of the corridor or by the lead scientist moving ahead to compel the ostrich to follow in our experiments. Tested running speed was lower than that ostrich ran for surviving in the field. In our study, the toe-joint motion only aimed at slow running gait instead of the fast running observed in the field. We think that it would be interesting to investigate the phalangeal joint kinematics at top running speeds.

We have several important statement about the number of experiment individuals. At first, five female and five male healthy sub-adult ostriches were chosen as the experiment individuals. Though they were sub-adult, we found that the male ostriches were very dangerous in the process of training. Thus, we decided to give up these male ostriches to do experiments. In addition, female ostriches were nature timid and afraid of researchers. On the runway 1.5 m high wire mesh fence, some female ostriches always crashed into the fence and hurt hindlimb in the process of running training. Finally, we had to choose two tractable female ostriches as experimental objects. We believe that results of phalangeal joint kinematics for only two individuals may still be valuable for ostrich toe study in the future.

Perspectives

The gait measurements in this study were conducted on solid level ground surface. Future work involves the investigation of the toe-joint motions when moving on rough terrain at slow and fast speeds close to 50 km/h, and also during sideways maneuvers. This would enable us to inspect gait motions and foot bio-mechanics of ostriches when moving in an experimental setup closer to their natural habitat. In addition, how the ostrich foot generates sufficient braking and propulsive forces on granular media is of high interest to us. Moreover, the driving mechanism of the ostrich toe complex with a unique supra-jointed posture might inspire development of innovative bipedal robots capable of running as fast and as economically as ostriches.

Conclusion

All the six major toe joints investigated in this study show noticeable motions from slow to fast gaits. The MTP3 and MTP4 joints present the largest range of motion whereas the first phalangeal joint of the 4th toe exhibits the largest motion variability. The interphalangeal joints of the 3rd and 4th toes show very similar motion patterns during slow walking and running. However, the MTP3 and MTP4 joints motions and also the vertical displacement of the metatarsophalangeal joint present significantly different patterns during running and slow walking. Because of biomechanical requirements, ostrich are likely to select the inverted pendulum gait at low speeds and also the bouncing gait at high speeds to improve movement performance and energy economy.

Indeed, the motions of the MTP3 and MTP4 joints are highly synchronized across the entire speed range examined in this study. This strongly indicates that the 3rd and 4th toes actually work as an “integrated system” with the 3rd toe as the primary load bearing element whilst the 4th toe as the complementary load sharing element mainly to ensure the lateral stability of the permanently elevated metatarsophalangeal joint.

Supplemental Information

Table S1 The six indicators (angles/displacements at touch-down, mid-stance, lift-off, maximum, minimum and range of motion) of the six toe joint angles and the vertical displacement of the metatarsophalangeal joint

Click here for additional data file.

Supplemental Information 1 Statistical data: mean force and s.d. with the variation of stance period

Click here for additional data file.

Additional Information and Declarations

Competing Interests

Author Contributions

Animal Ethics

Data Availability

The authors declare there are no competing interests.

Rui Zhang conceived and designed the experiments, wrote the paper, reviewed drafts of the paper.

Qiaoli Ji conceived and designed the experiments, analyzed the data, wrote the paper, prepared figures and/or tables.

Gang Luo, Shuliang Xue and Songsong Ma performed the experiments, analyzed the data, wrote the paper.

Jianqiao Li contributed reagents/materials/analysis tools, reviewed drafts of the paper.

Lei Ren conceived and designed the experiments, contributed reagents/materials/analysis tools, reviewed drafts of the paper.

The following information was supplied relating to ethical approvals (i.e., approving body and any reference numbers):

All living and experimental conditions were approved by the Institutional Animal Care and Use Committee (IACUC, protocol number: 20140706) of Jilin University, P.R. China.

The following information was supplied regarding data availability:

The raw data has been supplied as Data S1.

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
