# Peer review of "Phalangeal joints kinematics during ostrich (Struthio camelus) locomotion"

_PeerJ, doi:10.7717/peerj.2857_

## Round 0.1 · original submission · Major Revisions

· Academic Editor

Major Revisions

I have now received three reviews of your manuscript. All reviewers agree that this study is a good piece of work, however it needs a profound revision before it can be considered for publication. One main concern is your small sample size. I do not understand why did you trained ten specimens and then selected only two of them to do the trials. Apart from this, all three reviewers found several flaws in your experimental design. These flaws should be fixed in order to this manuscript can be accepted. Our second reviewer is particularly concerned with your writing, please take those suggestions into consideration.

I recommend to take account of all reviewer's suggestions, and make a point by point rebuttal letter.

·

Basic reporting

Overall the article did a good job of presenting background literature and how this study fits into the field. It is structured well and is contains an appropriate "unit of publication". It would make a great addition to the scientific literature and I believe it would be of great interest to those who study comparative biology, zoology, bio-inspired engineering, and other similar fields.

There were a few small changes related to the figures and captions which I would suggest to improve clarity:
In lines 203-206 (Figure 2), the letters of the angles referenced within the caption should match the greek letters in Figure 2C.
In lines 224-227 (Figure 3), To be consistent between the figure and caption, either the ms time references should be added to the figure or the caption should use % stance phase.
Line 281 (Figure 4): I think it would be clearer if the y-axis of graphs corresponding to the same angles were scaled equally in order to more easily see difference in magnitude.

Collection of pressure mat data was mentioned at the start of the "Perspectives" section. This data seems like it would be very relevant to the study, therefore it should be included or not mentioned.

Experimental design

The authors presented an original study which fills a very well defined gap in knowledge. Due to the challenging nature of measuring phalangeal joint kinematics, I do have a few concerns, primarily related to the accuracy and completeness of the experimental design:

Line 51-52: While the subjects were selected from a larger population for certain characteristics, the data presented is from 2 subjects, therefore I think the abstract should clearly state that n=2.

Line 181-183: It is unclear which feet were markered and used in the study - 1 foot of each ostrich? Always left/right?

Line 194-196: While I understand that the kinematics of all phalanges would be very hard to measure, the justification of why they were not measured is not satisfying. How small are their motions compared to the other joints? Fowler et al. 1991 did not describe comparative mobility of these joints. Is there any analytical explanation of why some joints were analyzed and not others?

Line 209: Since the accuracy of this 3-camera system used to reconstruct 3d motion is not publicly known (as a fully commercial motion capture system would be), the authors should provide some indication of system accuracy.

Line 229-234: If velocity is calculated from the MTP3 joint marker, it would represent the swinging of the leg as opposed to a more steady-state body velocity. It isn’t clear to me that two similar consecutive instantaneous velocities at touchdown would indicate that the ostrich center of mass (COM) is moving at a steady state. Additionally, I do not know how valid the Froude number would be if the velocity was not of the COM.

Validity of the findings

Overall, the presentation of the statistical analysis was thorough and clear. The only concern I have lies with the small sample size. When only two subjects are used, statistical findings could be viewed by some as invalid. Some more details about how the two subjects data compared could help validate these statistical methods. How many of the trials presented were contributed by each subject? which variables (if any) showed statistically significant differences between the two subjects?

Line 263 (Table 1): the average speed, froude number, and duty factor seem to be significantly different between walking and running, but are not denoted so in the table.

Additional comments

In the discussion, end of 1st paragraph: The two subjects chosen may be an OK sampling genetically since they are unrelated, but it is very limiting in terms of age, size, maturity, etc.

In the discussion, end of middle paragraph on pg 17: I would have guessed (pure speculation) that the major function of the 4th toe would be to provide stability in turning maneuvers. Do the authors not believe this to be the case?

The CT used for anatomical reference was older and heavier than the subject population. Are the authors aware of any differences which may exist between this reference and their younger experimental subjects?

Was the swing phase analyzed at all? It was included for calculation of gait parameters, so why was it omitted from this study?

I noticed a few spelling errors (e.g. line 86 "muscel-tendon") throughout the paper that the authors should fix before publication (though this has no effect on the scientific merit of the article).

Reviewer 2 ·

Basic reporting

1) I feel that the manuscript, in its current form is difficult to read. There are a number of specific reasons for this and I have provided them (with examples) here.
a. There are a large number of incomplete sentences. For example, Introduction, Pg 3, Line 88-89 reads “Furthermore, the ligaments system and tendon in or near the hindlimb joints have the important functions for ostrich economic and fast locomotion.” I believe a simple, direct complete sentence such as, “Furthermore, the ligaments and tendons in the distal joints likely play a role in ostrich gait performance and economy” captures the authors intent. Other examples of incomplete/unclear sentences can be found on (not a complete list):
i. Introduction, Pg. 3, Line 99-100
ii. Introduction, Pg. 4, Line 107-110
iii. Methods, Pg. 5, Line 144-147
iv. Methods, Pg. 8, Lines 240-242
v. Discussion, Pg. 17, first sentence (“During walking and running….”).
I recommend that the authors carefully check the entire manuscript to identify and correct this issue.
b. Many of the paragraphs within the manuscript lack a clear/strong topic sentence that helps to guide the reader through the paragraph. As a result, ascertaining what are the authors key points can be difficult to do. For example, the first paragraph of the introduction provides a list of ostrich gait characteristics that have been observed previously in other studies. An introductory sentence along the lines, “Ostriches have a large number of adaptations that allow them to move both economically and quickly.” Please check the manuscript so that the main points of each paragraph are clearly identified and introduced.
2) Introduction/Background
a. The authors use the terms “intertarsal” and “ankle” interchangeably throughout the manuscript. Please use just one term.
b. Intro, Line 86. The authors refer to the ostrich has having and erect femur. What is this compared to? Relative to humans, the ostrich femur is very flexed. I do not think that the femur is greatly contributing to the leg length when compared to other vertebrates.
c. Intro, Line 93-96: In this sentence and many others throughout the manuscript, the authors refer to the distal tendons as storing and releasing energy. While I agree this is likely the case, the current manuscript presents this as factual. However, this is only conjecture. Indeed, one of the main reasons that many of the previous studies exist is to try and confirm if this is the case! Please be careful to not state this concept as fact, but as a likely phenomenon.
d. Introduction, Lines 112-119. Here the authors state that previous studies have not focused on the MTP joint. I believe that this is incorrect. For example, recent studies such as Hutchinson et al (2015) and Rubenson et al (2010) that are cited in the manuscript investigate MTP joints. In addition, a more recent study by Rankin et al (Rankin, JW, Rubenson, J, Hutchinson JR. 2016. Insights gained from a three-dimension musculoskeletal model into ostrich pelvic limb muscle roles during walking and running gaits. Royal Society Interface. 13. 118) looks at MTP joint muscle function. However, all these studies were limited in their characterization of the foot in that it is represented as a single rigid body instead of individual toe joints and separate MTP3, MTP4 joints. I think this is an important point that needs to be clarified as this study goes beyond the current “state-of-the-art” by looking at interphalangeal motion.
e. The hypothesis, as stated are not clear to me. What is meant by “work as an integrated whole”? How is this measured? In both the introduction and the discussion this term was not clear to me.
f. Introduciton, Lines 135-137. The sentence beginning with “This study may further…” is redundant and should be removed.
3) Figures
a. Figure 2C: How was MTP height (z) measured? Please add to the schematic and discuss in the methods.
b. Figure 2C: Throughout the manuscript the authors use the greek symbols to refer to joints. It would be much better if the authors used actual joint names and then the symbols parenthetically. For example, the phrase “the angle a shared very similar” could be written as “the first phalangeal joint of toe IV (a) shared very similar…”. This would greatly improve the clarity of the discussion.
c. Figure 2, caption: Please ensure the Greek letters are used in the caption.
d. Figure 3, caption. Remove the sentence, “Time for each frame was indicated in milliseconds”. This does not pertain to the associated figure. I would put the midstance time values in the results, not the figure caption.
e. Figure 4: I suggest that the plots from walking and running are superimposed. This would greatly improve visual comparison. For example, Plots A and B could be on a single figure panel with walking data in one color and running data in a different color.
f. Figure 5, Table 2: These are redundant. I recommend placing the table in supplementary information.

Experimental design

I have only a few minor comments on experimental design.
1) The authors indicate that the running speed in their study was 2.77m/s. This is much lower than the values presented in the introduction 50km/h (~13 m/s). I believe the authors need to mention this as a limitation of their study, why the speed is so much lower and what effect, if any, it could have on study results.
2) In the results and discussion, the authors mention “key indicators”. These need to be clearly defined and related to their hypothesis. For example, a sentence such as, “To test our hypothesis that joint motions differed between the two toes, joint range of motion, maximum and minimum joint angles were selected as key indicators for statistical tests”.

Validity of the findings

Results
1) As mentioned above, please refer to your results using both the anatomical joint definition as well as the joint angle symbol.
2) In line 275, the authors state that one of the joints (beta?) experiences protraction. However, the joint angles presented in the methods are only flexion/extension angles. Please remove or correct.
Discussion
1) First paragraph. The authors state that two ostriches are representative of all ostrich locomotion. However the authors never provide any data for comparison. I would recommend providing comparison data as supplementary info. How did the authors select representative birds? How do their kinematics relate to a larger data set? If this cannot be provided, then the authors need to be careful in stating that the results of two birds (regardless of how many trials) can be generalized to the entire population.
2) Second paragraph. The last sentence beginning with “This suggests that the toes…” is an important conclusion of the study. I suggest expanding on this concept in a separate paragraph. For example how can the toes play and active role when there are no muscles? What are the mechanisms?
3) Page 17, second paragraph. The authors state “This appears to support the previous hypothesis…”. What hypothesis are the authors referring to? The current paper does not hypothesize anything about the role of the 4th toe as a stabilizer. Please clarify.
4) The last two sentences of the same paragraph seem contradictory. In the first the authors state that “the body stabilization function of the 4th toe…may not be as pronounced.” The next sentence then states this toe, “provides extra support for the lateral stability of the joint”. Which is it? Please clarify these sentences. I also suggest providing more detail from the referenced Schaller study. How do these results specifically compare to the findings of Schaller 2011. What are the differences? How might they be explained?
5) Top of page 19. The authors state “implying different neuromuscular control strategy is used compared to slow moving”. This statement appears to be purely conjecture and I would be very cautious in stating that changes in kinematics are due to changes in neuro control. The authors have not presented or reference any muscle activity data, so one can not actually say the changes are due to a change in neural control strategy. Instead it could be a biomechanical/passive consequence of using the same strategy for two different gaits/speeds. I recommend removing this statement completely.
6) Page 19, sentence beginning with “This is probably because of the constant leg length…”. I believe that this is difficult to ascertain from the current study results. The MTP height is a very small part of the overall leg length. Without knowing/presenting the kinematics of the rest of the limb, one cannot make this statement.
Acknowledgements
1) The acknowledgements section is not an acknowledgement. Instead it is a statement of contributions by the authors.

Additional comments

The authors prevent an interesting study that investigates individual toe joint kinematics in ostriches during walking and slow-running movements. The findings provide new information that add to a growing body of literature directed at improving our understanding of ostrich gait. I believe that the study results would be of interest to other researchers in this area. However, based on the current version of the manuscript, I have presented a number of questions and concerns that I believe should be addressed before publication.

Reviewer 3 ·

Basic reporting

This manuscript reports about Ostrich foot kinematics at lower and higher locomotion speeds. The manuscript is well written (as far as I can tell, being not a native speaker).
The topic is interesting and as far as I know not addressed before. The authors report that MPT3 and MTP4 exhibit the larger range of motion while the 4th toe larger motion variability. They also found that MTP3 and MTP4 differ during low from higher speeds. The authors wrote in their paper during walking from running. Here, I found my first problem with this manuscript. The authors use duty factor to separate walking from running. Duty factor can only separate gaits containing double support phases from others having aerial phases. In avian locomotion, the existence of grounded running (a bouncing gait with a duty factor above 0.5) oblige the use of other methods (e.g. % congruity) to separate gaits. In my opinion, the authors can talk about gaits at lower speeds or gaits with double support, but not about walking.

Experimental design

Here I found the most important flaw in the manuscript. The authors report first that they trained 10 healthy sub-adult ostriches, however for the locomotion analysis they only used two. In my opinion, without a strong argument (which is not given in the manuscript), two exemplars are not enough for a scientific report. Here I would recommend to include data from at least 2 or three more exemplars to the analyzed cohort.

Validity of the findings

Results are interesting but their validity is quite constrained due to the reduced numbers of analyzed subjects.
The conclusion is well written, however sometimes the authors tend to overinterpretate their results. The authors should include some relevant literature.

Additional comments

Specific comments:
From discussion the manuscript is not numbered. This makes the review more complicated!
Line 125 replace hapothesis by hypothesis.
Line 233 Froude number: The authors did not explain how they measured h (leg length?)
Line 261 “Interestingly, no statistically significant difference was found in swing duration between walking and running gaits.” This observation is no new! Please refer e.g. to following papers[1-4]
Line 338 please rewrite
Discussion lines starting with “This suggest that the toes might play an active…”. This sentence is too general and at the same time too speculative. Of course they play a role in force-generation and may be also in energy saving. However, without torque data combined with joint kinematics, you cannot say nothing about. Toe might also work as dampers!
Discussion lines starting with “this suggests that the tensions at the toe flexor tendons are well tuned by some of the tendons…” too general. Ether you name the structures that contributes to the observed behavior or you say nothing.
Discussion lines starting with “Slow moving may need a greater level of neural control and muscular…” here you need some references.
Discussion “The leg muscles of ostriches are highly concentrated at the proximal joints resulting in a low moment of inertia with respect to the proximal joints”. The word low is here maybe misleading. maybe reduced?
Discussion lines starting with “Indeed, the ostrich toes can be considered as a passive mechanism..” This is not correct; the foot is not passive! Foot’s joints are maybe underactuated but they are definitively not passive.

Discussion lines starting with “However, at high speeds, the metatarsophalangeal joint presents a typical loading and rebounding pattern…” that the metatarsophalangeal joint works like a spring was already described by [5].
Discussion last paragraph: There are a plenty of template related studies that describe the leg as an actuator in series with a spring to model e.g. avian locomotion. Please refer to that studies.

Perspectives lines starting with “Moreover, the tendon driven passive mechanism…” again if driven cannot be passive!

1. Nyakatura J.A., Andrada E., Grimm N., Weise H., Fischer M.S. 2012 Kinematics and Center of Mass Mechanics During Terrestrial Locomotion in Northern Lapwings (Vanellus vanellus, Charadriiformes). J Exp Zool Part A: Ecological Genetics and Physiology 317(9), 580-594. (doi:10.1002/jez.1750).
2. Andrada E., Nyakatura J.A., Bergmann F., Blickhan R. 2013 Adjustments of global and local hindlimb properties during terrestrial locomotion of the common quail (Coturnix coturnix). The Journal of Experimental Biology 216(Pt 20), 3906-3916.
3. Stoessel A., Fischer M.S. 2012 Comparative intralimb coordination in avian bipedal locomotion. The Journal of Experimental Biology 215(Pt 23), 4055-4069. (doi:10.1242/jeb.070458).
4. Kilbourne B.M., Andrada E., Fischer M.S., Nyakatura J.A. 2016 Morphology and motion: hindlimb proportions and swing phase kinematics in terrestrially locomoting charadriiform birds. Journal of Experimental Biology 219(9), 1405-1416.
5. Rubenson J., Lloyd D.G., Heliams D.B., Besier T.F., Fournier P.A. 2010 Adaptations for economical bipedal running: the effect of limb structure on three-dimensional joint mechanics. Journal of The Royal Society Interface 8(58), 740-755. (doi:10.1098/rsif.2010.0466).

---

## Round 0.2 · Major Revisions

· Academic Editor

Major Revisions

As you may see, there are still some issues identified by our reviewers that should be taken into consideration. Please, pay attention to the suggestion of our first reviewer in relation to the presentation of your data. I think that by follow one of those possibilities you will improve your manuscript considerably. Definition of some of the angles that you defined still needs clarification. Please take into account all suggestions exposed in this new round.

·

Basic reporting

I still see spelling and errors in sentence structure, despite the concerns expressed by multiple reviewers previously. Here is a non-exhaustive list:
Line 46: “The ostrich is highly cursorial bipedal…” should be “The ostrich is a highly cursorial bipedal…”
Line 86: “muscel” should be “muscle”
Line 95: “plays” should be “play”

Line 355 (Figure 5): This figure could be improved by somehow indicating which of the 28 comparisons between walking and running were significant. As it currently is presented, it is a little overwhelming and unclear what data is of greatest interest.

Experimental design

What surface was the data acquisition zone/ runway? You should include this in the experimental description especially since (as you mentioned in the discussion) another question of relevance is how kinematics change with different particle substrates.

I am still concerned with the accuracy of this system. There is obviously some sacrifice in quality from this simple system compared to using an 8-12 camera IR motion capture system. This can be simply tested by moving around a markered wand of known distance to calculate accuracy. An average or maximum error (e.g. +/- 0.5mm) should be presented.

Validity of the findings

My primary concern remains with the limitation of 2 subjects. While I find a two-subject study valuable, I believe the analysis needs to be structured accordingly (i.e. presented more like a case study than a sample representing a population). Combining the results of the two subjects could be masking or inversely, over-emphasizing certain results, especially with different amount of trials from each subject being included in the results. The way that the kinematics are being combined, it is unknown how much of the variation viewed in the results is due to intra-variability vs inter-variability. The Schaller (2011) pressure study cited by the authors does utilize a similar study design, however they show that they found no significant inter-individual variation (by performing an ANOVA). I would strongly suggest either of the two following actions be taken with regards to presenting data with two subjects:
Present all data from each subject separately (i.e. a double case study)
Present the data together, but include in the analysis equal numbers of trials from each subject for both walking and running (Otherwise one subject is being “weighted” in the average more than the other). Additionally, add inter-subject comparisons for all output variables.

Line 427: I think it is a possibility that this larger variance that is observed may be related to the fact that the distal segment of this joint is an approximation of phalanges II - V. There may be some smaller motions within this simplified segment as well as a high sensitivity to marker placement factoring into this observed variability.

Additional comments

I think some good improvements have been made to the article since the last submission. Unfortunately I don't find a couple of my greater concerns fully addressed yet.

Reviewer 2 ·

Basic reporting

The authors have done an admirable job in addressing my concerns and the manuscript is much improved. I have a few additional thoughts that I feel might further improve the presentation of this work.

1) Within the manuscript the authors have opted to use the phrase “gaits with double support” instead of the more common word “walking” based on the comments of reviewer three. The approach used in this study separated gaits based primarily on duty factor (>0.5 vs <0.5) and the authors mention in the rebuttal that the gaits presented were “slow walking gaits” and “running gaits”. While I agree ostriches do use a “grounded running” gait that you may have classified with the slower gaits, I personally believe that using the word “walking” instead of “gaits with double support” could still be an appropriate description of your data and make the manuscript easier to read. However, I will leave this to the discretion of the authors.

2) The new combined figure is much improved from the previous version. However, I recommend a few minor changes to the figure to further improve its ability to convey the important information
a. The figure needs a legend
b. The figure colours do not print well in black and white (maybe use a dotted/dashed line for one of the mean values?).
c. The labels for the different panels (joint names) could be bigger. Also, I recommend using “percent stance” instead of “proportion of stance period”
d. The addition of the kinematics is also very nice. I suggest adding the MTP location (green dots) to the running trial to better match the walking trial. Also I did not see where Figure 4H and Figure 4I were referenced in the manuscript.

3) Abstract, Line 39: please insert “a” between “is” and “cursorial”

4) Abstract, Line 51-51: The sentence beginning with “This is probably because...” is unclear. The sentence, as constructed, seems to indicate that ostriches are required to use an inverted pendulum gait and a bouncing gait. However, this is not necessarily the case. Instead it is likely ostriches have selected to utilize these types of gait to improve performance and conserve energy.

5) Introduction, Line 74-75: I suggest rephrasing the sentence “Their erect femurs” to state “Compared to other terrestrial birds such as rheas (Rhea spp.), emus (…) and brown kiwis (…), ostriches relatively erect femurs increase the joint chain extension and symmetrical movement…”.

6) Throughout the manuscript the authors use “MTP3”, “MTP4” and “metatarsophalangeal joint”. How “metatarsophalangeal joint” differs from MTP3, MTP4 is unclear. Please define in the manuscript.

7) Introduction, Line 108-109: The authors have continued to use the phrase “as an integrated whole” in their hypothesis. I still feel that the meaning of this phrase is not well defined prior to its use. The authors do an admirable job explaining what they mean in the Abstract (Lines 53-54, “This strongly suggests that the 3rd and 4th toes…”). I believe a similar sentence should be in the introduction before using the phrase “integrated whole”

8) I suggest using the word “individuals” instead of “specimens” when referring to the ostriches used in the study (e.g., Line 127).

9) Line 241: I believe that the word “second” should be “third” in the statement “The second phalangeal joint…”

10) Lines 331-335: The authors have added some statements regarding why sub-adult birds are representative based on previous work. The two sentences beginning with “While…” are fragments and need to be revised.

11) Line 336: “active” usually refers to muscle activity. I recommend replacing “active” with either “critical” or “essential”

12) Figure 5: Please add what the * means to the figure caption.

13) Lines 349-350: The wording of these two sentences beginning with “At slow speed state…” seem to indicate that the ostriches store energy while walking and then release the stored walking energy during running to be economical. I believe what the authors intended to say was that the MTP joint stores and returns more energy during faster (bouncing gaits) than slower gaits, which is a reasonable conjecture based on the study and previous research. Ostriches do not create a reservoir of energy during walking that is subsequently released during running.

Experimental design

1) Lines 182, 282-283: The authors have also done a good job clarifying their use of symbols for joint angles. However, I believe that their definition of their angle between the 3rd and 4th toes needs a slight clarification. I believe that this angle is meant to be the angle between the toes as viewed from above (i.e., how adducted toe 4 is relative to toe 3) as opposed to the difference in angle between MTP3 and MTP4 (i.e., sagittal plane). Please add a statement here (Line 182) clarifying the angle’s orientation as it is not clear in Figure 2 (which is 2D). This is further confused in lines 282-283 where the authors use flexion and extension.

2) Lines 444-448: The authors include here some statements about why their running values were lower than those observed in the field. However, they do not provide any reason why their results are still valid. Please add a sentence explaining why the slow running from the study can represent fast running observed in the field.

Validity of the findings

1) I agree with the two other reviewers comments that using only 2 individuals makes it difficult to obtain generalizable results, especially when 10 birds were available. I believe that the authors should have a paragraph in the discussion that addresses this limitation and why they chose to use 2 animals similar to the response provided to the reviewers.

2) The authors have provided the number of trials collected. However, the more relevant variable of interest is the total number of strides analyzed (i.e., how many stance phases were used in the data set?). Please add this to Table 1.

Additional comments

The authors have done an admirable job in addressing my concerns and the manuscript is much improved. I have a few additional thoughts that I feel might further improve the presentation of this work.

Reviewer 3 ·

Basic reporting

I'm mostly happy with the changes made by the authors. They have improved the manuscript as far as possible, given the reduced number of specimens. More individuals would certainly have increased the impact of the present study. Still, results are of interest for the community and should be publish..

Experimental design

No Coments

Validity of the findings

No Comments

Additional comments

I'm mostly happy with the changes made by the authors.
There are a lot of typos’ errors in the new added sentences. Please double-check.

---

## Round 0.3 · Minor Revisions

· Academic Editor

Minor Revisions

I think that we are almost ready to finish this review process. Please, pay attention to the suggestions of our reviewer.

·

Basic reporting

There remains some sentences which are written with poor English. A native English speaker correcting these sentences will improve clarity.

Fig6 caption: you can just say no significant effects were found, since there are no * in the figure).

Experimental design

The additional description of the angle between the 3rd and 4th toes confused me more. I previously thought that the angle was the 3D angle of the two fist phalanges relative to each other. With the added description and image indicating that the angle is "viewed from above" it now seems that my previous understanding was incorrect and the angle is actually a projection of the joint onto a global transverse plane. Visually, the angle defined in Fig2 C and D seem to indicate contradictory definitions.

I appreciate your sharing the updated stance phase counts for each subject with us, however I think this should be worked into the paper somewhere as well so that the readers are reassured that that both subjects were weighted equally (either Statistical Analysis section or Results:Gait Parameters?). Something like "An equal number of strides were included in the statistical analysis from each subject for both running and walking in order to weight each evenly" or including the actual stride breakdown like in the rebuttal letter would be sufficient.

Validity of the findings

No Comments

Additional comments

My concerns from previous reviews have largely been addressed and believe the manuscript, with a few minor clarifications, to be a valuable addition to the field.

---

## Round 0.4 · accepted · Accept

· Academic Editor

Accept

I am happy to see that the hard work of both, reviewers and authors, ended in this nice manuscript.
I attached the v3 of the manuscript again because I noted that some language issues persist. Adittionally, in Figure 2 a space should be inserted between toe and nail. Please, make an effort to assure that our international audience will clearly understand your text.